# Chrysin Protects against Memory and Hippocampal Neurogenesis Depletion in D-Galactose-Induced Aging in Rats

**DOI:** 10.3390/nu12041100

**Published:** 2020-04-16

**Authors:** Ram Prajit, Nataya Sritawan, Kornrawee Suwannakot, Salinee Naewla, Anusara Aranarochana, Apiwat Sirichoat, Wanassanan Pannangrong, Peter Wigmore, Jariya Umka Welbat

**Affiliations:** 1Department of Anatomy, Faculty of Medicine, Khon Kaen University, Khon Kaen 40002, Thailand; rooropt@gmail.com (R.P.); nataya117@hotmail.com (N.S.); kornraweeann@gmail.com (K.S.); salinee_rtu@yahoo.com (S.N.); anusar@kku.ac.th (A.A.); apiwsi@kku.ac.th (A.S.); wankun@kku.ac.th (W.P.); 2School of Life Sciences, Medical School, Queen’s Medical Centre, The University of Nottingham, Nottingham NG7 2RD, UK; peter.wigmore@nottingham.ac.uk; 3Neuroscience Research and Development Group, Khon Kaen University, Khon Kaen 40002, Thailand

**Keywords:** neurogenesis, aging, D-galactose, chrysin

## Abstract

The interruption of hippocampal neurogenesis due to aging impairs memory. The accumulation of D-galactose (D-gal), a monosaccharide, induces brain aging by causing oxidative stress and inflammation, resulting in neuronal cell damage and memory loss. Chrysin, an extracted flavonoid, has neuroprotective effects on memory. The present study aimed to investigate the effect of chrysin on memory and hippocampal neurogenesis in brains aged using D-gal. Male Sprague-Dawley rats received either D-gal (50 mg/kg) by i.p. injection, chrysin (10 or 30 mg/kg) by oral gavage, or D-gal (50 mg/kg) and chrysin (10 or 30 mg/kg) for 8 weeks. Memory was evaluated using novel object location (NOL) and novel object recognition (NOR) tests. Hippocampal neurogenesis was evaluated using Ki-67, 5-bromo-2′-deoxyuridine (BrdU), and doublecortin (DCX) immunofluorescence staining to determine cell proliferation, cell survival, and number of immature neurons, respectively. We found that D-gal administration resulted in memory impairment as measured by NOL and NOR tests and in depletions in cell proliferation, cell survival, and immature neurons. However, co-treatment with chrysin (10 or 30 mg/kg) attenuated these impairments. These results suggest that chrysin could potentially minimize memory and hippocampal neurogenesis depletions brought on by aging.

## 1. Introduction

Self-renewal is an important property of neural stem cells (NSCs) in neurogenic niches including the subventricular zone (SVZ) and the subgranular zone (SGZ). This phenomenon is potentially important in neurogenesis [1]. NSCs are involved in both the proliferation and differentiation of neurons and neuroglia. Newly generated neurons in the SGZ of hippocampal dentate gyrus (DG) are connected to the existing hippocampal circuit and play a role in memory [2,3].

The number of elderly people is currently predicted to increase by 21.5% by 2050 [4], and it is well-known that aging can lead to various chronic diseases [5]. The biomolecular mechanism of aging is associated with homeostasis disruption, which causes cell death [6]. Disturbing neurogenesis in the SGZ also causes memory impairment which is a major problem in brain aging [7,8].

D-galactose (D-gal) is found in milk, fruits, and vegetables [4]. It is a monosaccharide that is metabolized to glucose by galactokinase and uridyl transferase, which transports into the glycolysis pathway and gives energy to cells [4]. The accumulation of D-gal is oxidized to free radicals and then converted to galactitol or it interacts with amino proteins, which induces oxidative stress and inflammation and causes neuronal cell death [4,9,10]. For this reason, D-gal-induced cell death may lead to brain aging and is thus widely used in models of this process [11,12,13,14]. It has been demonstrated the effects of hesperidin (a flavonoid extracted from rind of oranges) on cognitive and mitochondrial dysfunction and apoptosis caused by D-gal-induced brain aging. The results reveal that the cognitive dysfunction in aging rats induced by D-gal is ameliorated by hesperidin using the Morris water maze test. Furthermore, the animals that received D-gal showed mitochondrial dysfunction by decreasing enzyme activities of the Kreb’s cycle and electron transport system and upregulating of caspase-3, marker of apoptosis. Co-administration with hesperidin and D-gal attenuates neuronal cell death in the CA1 [15].

Chrysin (5,7-dihydroxyflavone) is a flavonoid extracted from honey, propolis, and passion flowers [16,17,18]. Chrysin is used to treat liver, neurodegenerative, and reproductive system diseases [18,19,20,21]. Chrysin improves memory in the aging brain by attenuating increases in reactive species [18]. Chrysin also protects against hippocampal neuronal cell damage and improves memory deficits in rats suffering from chronic cerebral hypoperfusion [22]. Chrysin has the ability to diminish apoptosis and memory deficits caused by traumatic brain injury [23]. Moreover, the protective effects of chrysin on oxidative stress in D-galactose-induced aging rats have been demonstrated in a rat model. The data show that co-treatment with D-gal (50 mg/kg) and chrysin (20 mg/kg) by oral gavage for 8 weeks increases antioxidant enzyme activities and decreases MDA levels. Chrysin also ameliorates the toxicity of D-gal in brain tissue [16]. In a rat model, it has been reported that chrysin has high potential to protect against 3-Nitropropionic acid-induced mitochondria damage, oxidative stress, neuronal cell death, motor and cognitive impairments [24]. Chrysin also decreases impairments of the motor behavior, reduction of nigrostriatal dopaminergic neurons in retonone-induced Parkinson in rats [25].

This study focused on the effect of chrysin on memory and neurogenesis in rats that underwent D-gal-induced brain aging. Memory was evaluated using novel object location (NOL) and novel object recognition (NOR) tests. In addition, neurogenesis in the SGZ was determined using Ki-67, 5-bromo-2′-deoxyuridine (BrdU), and doublecortin (DCX) staining, which were used to examine cell proliferation, cell survival, and presence of immature neurons, respectively.

## 2. Materials and Methods

### 2.1. Animals

Male Sprague Dawley rats (age: 8 weeks, weight: 280–300 g) were purchased from Nomura Siam International Co., Ltd. (Bangkok, Thailand). The animals were kept in cages (3–4 animals/cage) under a 12-h light/12-h dark cycle and maintained at 23–25 °C with accessible food and water. Four weeks after arrival, they were randomly divided into six groups (11 animals/group): vehicle, D-gal, chrysin 10, chrysin 30, D-gal + chrysin 10, and D-gal + chrysin 30. The experimental procedure was approved by the Khon Kean University Ethics Committee in Animal Research (IACUC-KKU-22/61).

### 2.2. Drug Administration

The animals in the vehicle group received 1 mL/kg/day of propylene glycol (Ajax Finechem Pty Ltd., Auckland, New Zealand) by oral gavage and 1 mL/kg/day of 0.9% normal saline solution by intraperitoneal (i.p.) injection. The animals in the D-gal group were given 50 mg/kg/day of D-gal (Sigma Aldrich, St. Louis, MO, USA) dissolved in 0.9% normal saline solution by i.p. injection. The animals in the chrysin 10 and 30 groups were administered with chrysin, which is 97% purity determined by HPLC analysis, and purchased from Sigma Aldrich, St. Louis, MO, USA, at 10 or 30 mg/kg/day dissolved in propylene glycol by oral gavage. The animals in the D-gal + chrysin 10 and D-gal + chrysin 30 groups received D-gal at a similar dose to the D-gal group and chrysin at equivocal doses as the chrysin 10 and 30 groups, respectively. All drugs were administered for 8 weeks. The animals were intraperitoneally injected with BrdU (100 mg/kg/day, Sigma Aldrich, St. Louis, MO, USA) dissolved in 0.9% normal saline solution at the first, second, and third day of drug administration.

### 2.3. Behavioral Tests

Three days after drug administration, animals were subjected to the NOL test. First, they were habituated in an empty arena (width × length × height = 50 × 50 × 50 cm) for 30 min. One day later, in the familiarization trail, the animals freely explored two identical objects placed at individual locations of the arena for 3 min. Then, they were taken back to their cage for 15 min. Concurrently, olfactory cues were removed by cleaning the arena and objects with 20% ethanol. One of the objects was relocated to a novel location (NL), while the other one was placed in the familiar location (FL). In the choice trial, the animals were allowed to explore the objects for 3 min. The NOR test was performed one day after the NOL test. Habituation was conducted for 30 min. The next day, in the familiarization trail, two duplicate objects were located at different corners of the arena. The animals were then placed into the arena and allowed to explore the objects for 3 min, after which they were returned to their cage for 15 min. In the meantime, the arena and objects were cleaned with 20% ethanol. Then, a novel object (NO) and one of the familiar objects (FOs) were put into the arena. In the choice trail, the animals were allowed to explore the objects for 3 min [26,27].

The exploration time and movement in both tests were recorded using video tracking software of EthoVision^®^ XT (EthoVision^®^, XT version 12, Noldus, Wageningen, The Netherlands). The velocity and distance moved were tracked during the habituation phase. Exploration time was calculated as the time the rat spent with its nose directed at the objects at a distance of 2 cm or less. A preference index (PI) was calculated as the exploration time of the NL or NO in the choice trial as a percentage of the total exploration time compared to 50% chance [28,29,30].

### 2.4. Tissue Preparation

After the behavioral tests, the animals were euthanized, and cervical dislocation was performed. The brains were removed, separated into two hemispheres, and immersed in 30% sucrose for 3 h at 4 °C. They were then embedded in optimal cutting temperature (OCT) compound (Themo Fisher Scientific, Karlsruhe, Germany), frozen immediately in liquid nitrogen cooled isopentane, and stored at −80 °C for immunohistochemistry.

### 2.5. Immunohistochemistry

Frozen brains were cut serially throughout the length of the DG (Bregma −2.3 to −6.3) along the coronal plane using a cryostat (Cryostat Series HM 550 Microm international, A.S. Science. Co., Ltd., Walldorf, Germany). Sections for cell proliferation examination were cut at 20 µm thickness and examined using Ki-67 immunostaining. Nine sections per brain were selected from every 15 sections using a systemic random sampling technique [28,29,30]. The sections were incubated with primary antibody (Monoclonal mouse Ki-67, 1:150, Novocastra, Milton Keynes, UK) for 60 min and then secondary antibody Alexa Fluor 488 (1:300, Invitrogen, Eugene, OR, USA) for 40 min. Eventually, the sections were counter-stained with propidium iodide (1:6000, Sigma Aldrich, St. Louis, MO, USA) for 30 s. BrdU and DCX immunostaining were performed on 40 µm thick sections to determine cell survival and number of immature neurons, respectively. Nine sections per brain were selected from every 8 sections of the total DG [28,29,30]. For BrdU immunostaining, the sections were incubated with primary antibody (Anti-BrdU antibody, 1:100, Abcam, Cambridge, UK) at 4 °C overnight. They were then incubated with Alexa Flour 568 (1:300, Invitrogen, Eugene, OR, USA) for 60 min and counter-stained with diamidinophenylindole (DAPI, 1:6000, Molecularprobes, Eugene, OR, USA) for 30 s. In order to examine immature neurons, the sections were incubated with primary antibody (Doublecortin antibody, 1:100, Santa Cruz, Dallas, TX, USA) at 4 °C overnight followed by Alexa Fluor 488 (1:300, Invitrogen, Eugene, OR, USA) for 60 min. Finally, the sections were counter-stained with propidium iodide (1:6000, Sigma Aldrich, St. Louis, MO, USA) for 30 s.

### 2.6. Microscopic Quantification

The total numbers of Ki-67-, BrdU-, and DCX-positive cells within three cell diameters of the inner border of the DG were calculated from 9 coronal sections per brain using a 40X objective of fluorescence microscope (Nikon ECLIPSE 80i, Melville, NY, USA). The number of Ki-67-positive cells was multiplied by 15, and those of BrdU- and DCX-positive cells were multiplied by 8 [28,29,30].

### 2.7. Statistical Analysis

All statistical analysis was conducted using GraphPad Prism (Version 5.0, GraphPad Software Inc., San Diego, CA, USA) and presented as mean ± SEM. Significance was determined as *p* < 0.05. Exploration time was analyzed using a paired Student’s *t*-test. One-way repeated measure ANOVA was used to evaluate the movement and numbers of Ki-67-, BrdU-, and DCX-positive cells. Preference index was determined using a one-sample *t*-test.

## 3. Results

### 3.1. Effects of D-Gal and Chrysin on NOL Test Results

In the habituation trial, there were no significant differences in terms of distance moved (F5,35 = 1.702, *p* > 0.05, Table 1) or velocity (F5,35 = 1.698, *p* > 0.05, Table 1) among the groups. This study showed no differences in locomotor activity after receiving D-gal and chrysin. In the familiarization trial, animals in all groups spent an equal amount of time exploring the objects in locations A and B (*p* > 0.05, Figure 1A). In the choice trial, the animals in the vehicle, chrysin 10, chrysin 30, and D-gal + chrysin groups spent significantly longer exploring the object in the novel location than that in the familiar location (* *p* < 0.05, Figure 1B), but this was not observed in the D-gal group (*p* > 0.05, Figure 1B). These results suggest that D-gal impaired spatial memory, but that this impairment was mitigated by treatment with either 10 or 30 mg/kg of chrysin. In addition, the PIs of the vehicle, chrysin 10, chrysin 30, and D-gal + chrysin groups were significantly greater than 50% probability (vehicle group: * *p* < 0.05, chrysin 10 group: * *p* < 0.05, chrysin 30 group: * *p* < 0.05, D-gal + chrysin 10 group: ** *p* < 0.01, D-gal + chrysin 30 group: * *p* < 0.05, Figure 2), but that of the D-gal group was not (*p* > 0.05, Figure 2). These results demonstrate that D-gal induced spatial memory deficits. By contrast, spatial memory deficits were attenuated in the animals that received D-gal and either 10 or 30 mg/kg of chrysin.

### 3.2. Effects of D-Gal and Chrysin on NOR Test Results

Locomotor activity in the NOR test did not differ significantly among groups in terms of either distance moved (F5,35 = 1.031, *p* > 0.05, Table 2) or velocity (F5,35 = 1.036, *p* > 0.05, Table 2), indicating that D-gal and chrysin have no effect on locomotor activity. The exploration time of object A in the familiarization trial was similar to that of object B in all groups (*p* > 0.05, Figure 3A). In the choice trial, the exploration times of the novel object in the vehicle, chrysin 10, chrysin 30, and D-gal + chrysin groups were significantly longer than those of the familiar object (vehicle group: * *p* < 0.05, chrysin 10 group: *** *p* < 0.001, chrysin 30 group: ** *p* < 0.01, D-gal + chrysin 10 group: * *p* < 0.05, D-gal + chrysin 30 group: * *p* < 0.05, Figure 3B). By contrast, the animals in the D-gal group explored both objects for an equal amount of time (*p* > 0.05, Figure 3B), meaning that D-gal was the cause of recognition memory impairment and that chrysin at either 10 or 30 mg/kg was able to attenuate this impairment. The PIs of the vehicle, chrysin 10, chrysin 30, and D-gal + chrysin groups differed significantly from 50% chance (vehicle group: * *p* < 0.05, chrysin 10 group: *** *p* < 0.001, chrysin 30 group: ** *p* < 0.01, D-gal + chrysin 10 group: * *p* < 0.05, D-gal + chrysin 30 group: * *p* < 0.05, Figure 4), but that of the D-gal group did not (*p* > 0.05, Figure 4). This shows that the animals that received D-gal developed recognition memory impairment, but that it was reversed with chrysin administration at either 10 or 30 mg/kg.

### 3.3. Effects of D-Gal and Chrysin on Cell Proliferation in the SGZ

Ki-67 protein is expressed throughout the cell cycle except in the resting phase. This study used Ki-67 immunostaining to quantify cell proliferation in the SGZ (Figure 5A–F). Cell proliferation differed significantly among most of the groups (F5,30 = 17.09, *** *p* < 0.001, Figure 5G), but was similar in the chrysin administration and the vehicle group (*p* > 0.05, Figure 5G). By contrast, the number of proliferating cells in the D-gal group was significantly lower than in the vehicle group and both chrysin groups (*** *p* < 0.001, Figure 5G), indicating that D-gal had a negative effect on cell proliferation in the SGZ. However, the numbers of proliferating cells in both groups co-treated with chrysin were significantly greater than that of the D-gal group (*** *p* < 0.001, Figure 5G) and did not differ significantly from that of the vehicle group (*p* > 0.05, Figure 5G). This shows that chrysin at either 10 or 30 mg/kg was able to attenuate D-gal-induced cell proliferation decline in the SGZ.

### 3.4. Effects of D-Gal and Chrysin on Cell Survival in the SGZ

BrdU, a thymidine analogue, is an agent that can replace DNA during the S phase of the cell cycle. Accordingly, BrdU was injected into the animals at the beginning of this experiment to detect surviving cells in the SGZ (Figure 6A–F). There was significant variation in the number BrdU-positive cells among the groups (F5,30 = 55.24, *** *p* < 0.001, Figure 6G). Although the BrdU-positive cell numbers in the chrysin and the vehicle groups did not differ significantly (*p* > 0.05, Figure 6G), those in the D-gal groups were significantly lower (*** *p* < 0.001, Figure 6G). This suggests that D-gal reduced cell survival in the SGZ, but that this damage was significantly attenuated in both groups co-treated with chrysin (10 and 30 mg/kg; *** *p* < 0.001, Figure 6G), meaning that chrysin was effective in protecting against the cell death in the SGZ caused by D-gal.

### 3.5. Effects of D-Gal and Chrysin on Immature Neurons in the SGZ

Doublecortin (DCX) is a cytoskeletal protein that influences neuronal cell migration. Because this protein is mainly found in immature neurons, it was used to quantify them in this study (Figure 7A–F). DCX-positive cell count varied significantly among the groups (F5,30 = 61.29, *** *p* < 0.001, Figure 7G). Although it was similar in the vehicle and both chrysin groups (*p* > 0.05, Figure 7G), it was significantly lower in the D-gal group (*** *p* < 0.001, Figure 7G). Furthermore, there were significantly greater numbers of DCX-positive cells in the D-gal + chrysin groups than in the D-gal group (*** *p* < 0.001, Figure 7G). This suggests that chrysin (10 and 30 mg/kg) was effective in ameliorating immature neuron depletion in the SGZ caused by D-gal.

## 4. Discussion

This study demonstrated that chrysin protects against memory impairments resulting from D-gal-induced reduction of hippocampal neurogenesis. Animals that received D-gal exhibited memory impairments and disruptive neurogenesis, but co-treatment with chrysin (10 and 30 mg/kg) attenuated these impairments. We also found no significant differences in terms of distance moved or velocity among the groups, meaning that none of the animals were physically impaired in performing the behavioral tests.

In this study, NOL and NOR tests were used to evaluate spatial and recognition memory, respectively. Spatial memory is used to retrieve locations, configurations, or routes and depends on hippocampal function [31,32]. This memory is motivated in animals to find location of food or water, which is a tendency of animals to exhibit under natural conditions [33]. Recognition memory involves the recollection of experiences or events in the past [34,35] and depends on coordinated function between the hippocampus and cortical areas [36,37].

Various studies have found that D-gal-induced neuronal cell damage in the hippocampus impairs spatial memory [14,32]. In the NOL test, we imitated natural preferences of animals by changing the location of the object in the choice trial [33], and found this to be the case. Chrysin (10 and 30 mg/kg) had neither a negative nor a positive effect on spatial memory when compared to the vehicle group. Similarly, a previous study similarly found that chrysin at 50 mg/kg did not negatively affect spatial memory or numbers neuronal cells in hippocampus [38]. Moreover, we found that chrysin at 10 and 30 mg/kg was able to attenuate spatial memory impairment. This is consistent with the results of a previous study of the effects of chrysin (10 mg/kg) on brain aging, which found that it inhibits the down-regulating of antioxidant enzyme activities, sodium-potassium ATPase activity, and brain-derived neurotrophic factor and attenuates spatial memory impairment [18]. Moreover, chrysin (30 mg/kg) has been shown to restore spatial memory by protecting neuronal cell death in the DG in chronic cerebral hypoperfused rats [22].

Recognition memory was measured using the NOR test by replacing one of the familiar objects with a novel object. The animals in the D-gal group exhibited disruption of recognition memory. Our results correspond with those of previous brain aging studies, which have found recognition memory impairment in D-gal-induced hippocampal damage [32,39]. In both of the chrysin groups, recognition memory was similar to that in the vehicle group. Moreover, co-treatment with chrysin at doses of both 10 and 30 mg/kg attenuated recognition memory impairment. This is similar to the results of a previous study, in which animals that received chrysin at 50 mg/kg had no neuronal cell degeneration in the hippocampus or recognition memory impairment and that chrysin at both 25 and 50 mg/kg ameliorated recognition memory impairment rats with Alzheimer’s disease [38]. Chrysin (20 mg/kg) has also been found to protect against neuronal cell damage in aging rat brains [16]. Taken together, these results suggest that chrysin protects against hippocampal damage and attenuates reductions in spatial and recognition memory.

Neurogenesis occurs in the SGZ of the DG and plays an essential role in memory [1,3]. This phenomenon is associated with self-renewal of neural stem cells (NSCs) and new mature neurons in the SGZ [2,40]. This study utilized Ki-67, BrdU, and DCX to evaluate various phases of neurogenesis (cell proliferation, cell survival, and immature neurons, respectively; [28,29,30]. Levels of Ki-67 increase during all phases of cell division except the G0 phase, indicating that this protein is related to cell proliferation [41,42]. BrdU, a thymidine analogue, can merge into DNA during the S phase of cell division [43]. In this study, we traced the survival phase of the cells by injecting BrdU at the beginning of the procedure [28,29,30]. DCX is a protein that controls microtubules and actin skeleton in neuronal cell migration [44,45] and is only expressed in immature neurons [46].

The present study demonstrates that D-gal decreases neurogenesis, as indicated by decreases in the numbers of Ki-67-, BrdU-, and DCX-positive cells. Several studies have reported that D-gal interrupts neurogenesis by increasing oxidative stress and inflammation [13,14]. According to one study, D-gal stimulates lipid peroxidation that destroys Ki-67 and DCX [13]. D-gal impaired antioxidant enzyme activities and increased pro-inflammatory cytokines also decreases cell proliferation [14]. Moreover, a study of neuronal cell survival using BrdU and NeuN double-staining reveal that D-gal decreases neuronal cell survival by inducing oxidative damage and inflammation [47]. In this study, neurogenesis in the group given chrysin at 10 and 30 mg/kg did not differ from that in the vehicle group. Previous studies have reported that chrysin at both 20 and 50 mg/kg does not affect antioxidant enzyme activity levels [16,38]. This indicates that chrysin on its own does not have any benefit in terms of neurogenesis in adult rats. Furthermore, this study found that co-treatment with chrysin at either 10 or 30 mg/kg was effective in attenuating neurogenesis impairment caused by D-gal as indicated by cell proliferation, cell survival, and number of immature neurons. Although there is no evidence that chrysin enhances neurogenesis, it is able to protect against neuronal cell damage by inhibiting oxidative stress and inflammation [18,22,48]. In addition, chrysin can protect against neuronal cell death in the dentate gyrus by inhibiting MDA production in a chronic cerebral hypoperfused rat model [22]. It has been reported that the antioxidant and anti-inflammatory properties of chrysin ameliorates neuronal cell damage in cerebral ischemic-reperfusion injury mice [48]. Chrysin also improves antioxidant enzyme activities, brain-derived neurotrophic factor (BDNF) and memory in aging mice [18]. Chrysin also has antioxidant effects on liver disease in rats. [20]. Noticeably, the hydroxylation in the B ring of chrysin plays an important role in its antioxidant effect in rodents with liver disease [20]. Accordingly, chrysin can protect against reductions in cell proliferation, cell survival, and immature neurons caused by D-gal-induced aging. Several studies have proved that antioxidant and anti-inflammatory properties of chrysin are able to protect against neuronal cell damage and cognitive impairments. These properties would be a powerful effect of chrysin to ameliorate reduction in neurogenesis in aging rats caused by D-gal. Therefore, the mechanism of action of chrysin in aging rats including antioxidants and neurotrophic factors should be determined to confirm the biological effects of chrysin.

## 5. Conclusions

The present study demonstrates that D-gal is a cause of spatial and recognition memory impairment due to reductions in cell proliferation, cell survival, and immature neurons during the process of neurogenesis. Chrysin, while not able to enhance memory on its own, attenuated these deficiencies in adult rats.

## Figures and Tables

**Figure 1 nutrients-12-01100-f001:**
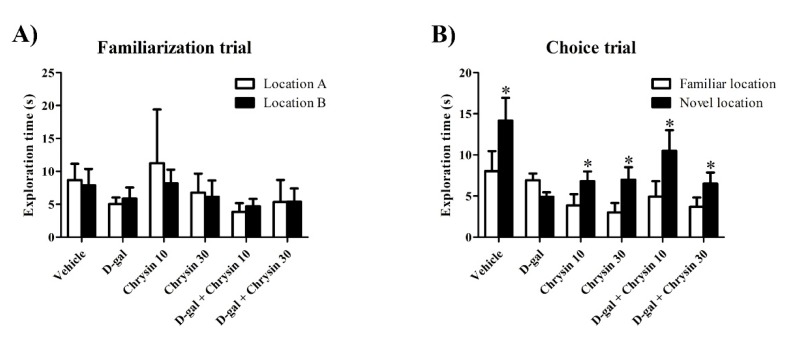
The exploration time (mean ± SEM) for each object in the NOL test after treatment. In the familiarization trial, no significant differences in exploration time between the objects in the two locations were found in any of the groups (*p* > 0.05, (**A**)). In the choice trial, the vehicle, chrysin 10, chrysin 30, and D-gal + chrysin groups explored the object in the novel location significantly longer than that in the familiar location (* *p* < 0.05, (**B**)), but those in the D-gal group did not.

**Figure 2 nutrients-12-01100-f002:**
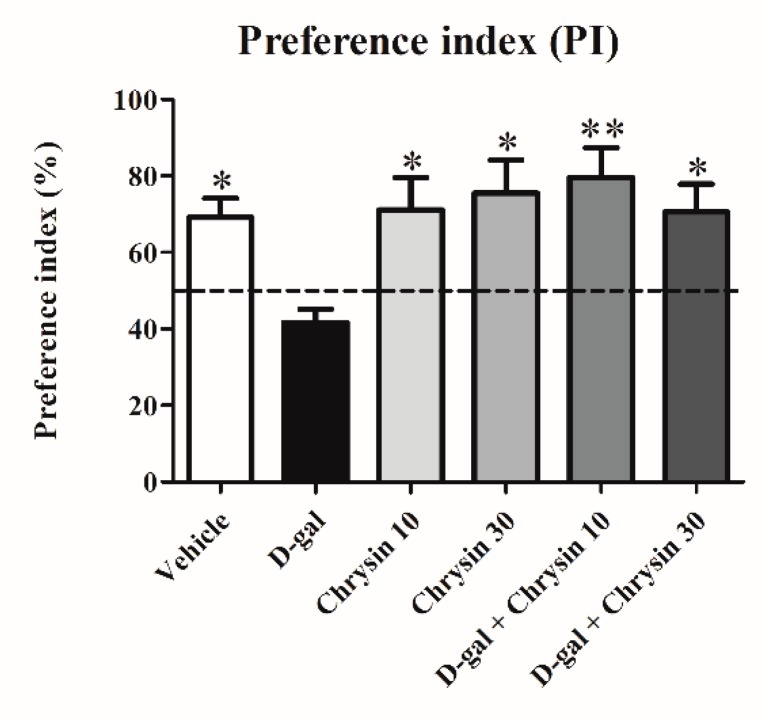
The preference indices (PIs, mean ± SEM) of the NOL test after treatment. The PIs of the vehicle, chrysin 10, chrysin 30, and D-gal + chrysin groups differed significantly from 50% chance (* *p* < 0.05, ** *p* < 0.01), but that in the D-gal group did not (*p* > 0.05).

**Figure 3 nutrients-12-01100-f003:**
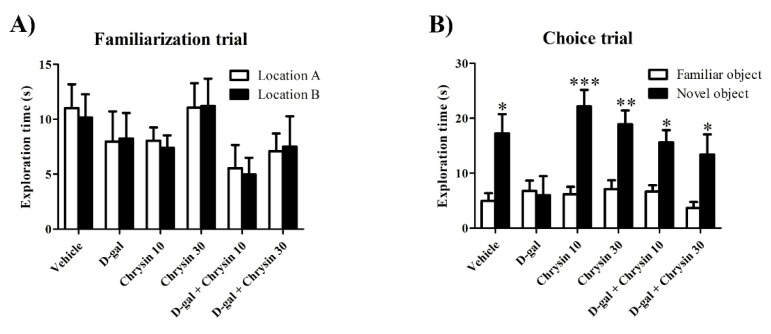
The exploration time (mean ± SEM) of each object in the NOR test after treatment. In the familiarization trial, there was no significant difference between objects A and B in terms of exploration time in any of the groups (*p* > 0.05, (**A**)). In the choice trial, the vehicle, chrysin 10, chrysin 30, and D-gal + chrysin groups explored the novel object significantly longer than the familiar object (* *p* < 0.05, ** *p* < 0.01, *** *p* < 0.001, (**B**)), but those in the D-gal group did not (*p* > 0.05, (**B**)).

**Figure 4 nutrients-12-01100-f004:**
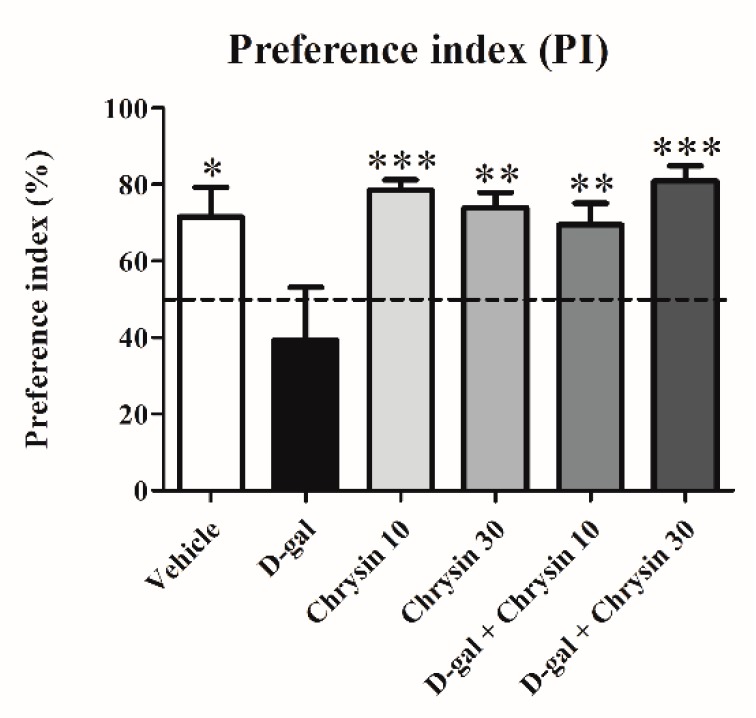
The PIs (mean ± SEM) of the NOR test after treatment. The PIs of the vehicle, chrysin 10, chrysin 30, and D-gal + chrysin groups differed significantly from 50% chance (* *p* < 0.05, ** *p* < 0.01, *** *p* < 0.001), but that of the D-gal group did not (*p* > 0.05).

**Figure 5 nutrients-12-01100-f005:**
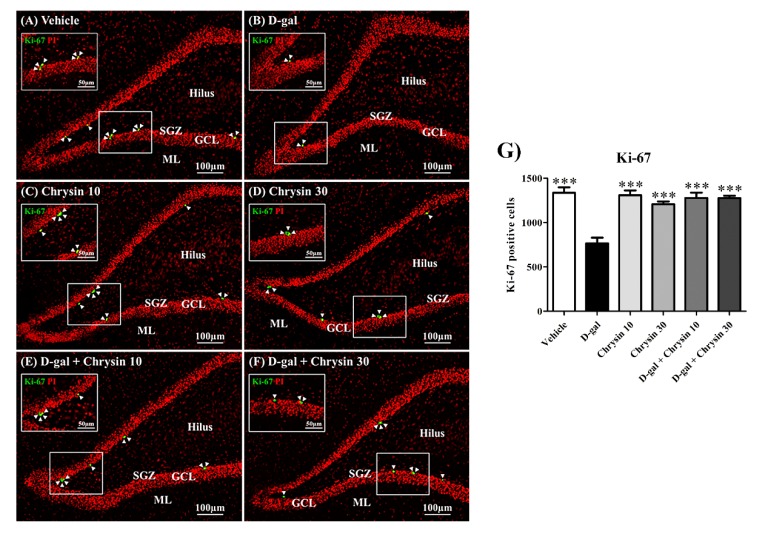
Immunofluorescence staining of all the experimental groups (**A**–**F**). Ki-67 positive cells (green) are indicated by arrowheads, and all nuclei were stained with propidium iodide (red). The vehicle, chrysin 10, chrysin 30, and D-gal + chrysin groups had significantly greater numbers of cells stained with Ki-67 than the D-gal group (*** *p* < 0.001, **G**). SGZ: subgranular zone; GCL: granule cell layer; ML: molecular layer.

**Figure 6 nutrients-12-01100-f006:**
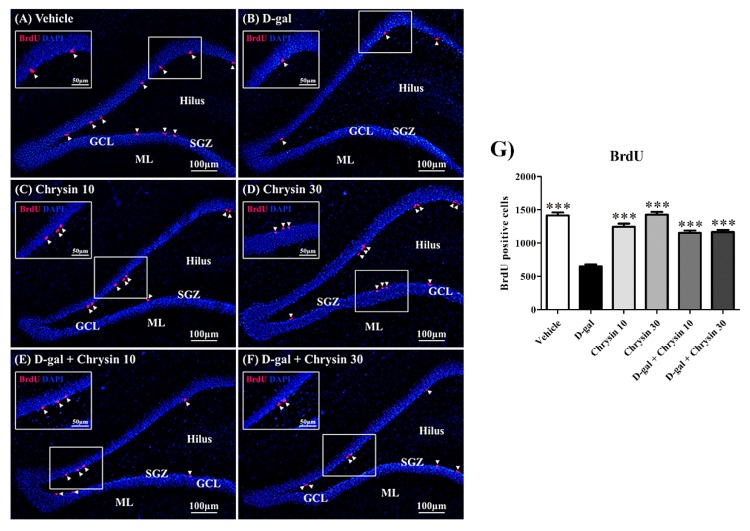
Immunofluorescence staining of all groups (**A**–**F**). BrdU positive cells (red) are indicated by arrowheads, and all nuclei were counterstained with DAPI (blue). The vehicle, chrysin 10, chrysin 30, and D-gal + chrysin groups had significantly greater numbers of cells stained with BrdU than the D-gal group (*** *p* < 0.001, (**G**)). SGZ: subgranular zone; GCL: granule cell layer; ML: molecular layer.

**Figure 7 nutrients-12-01100-f007:**
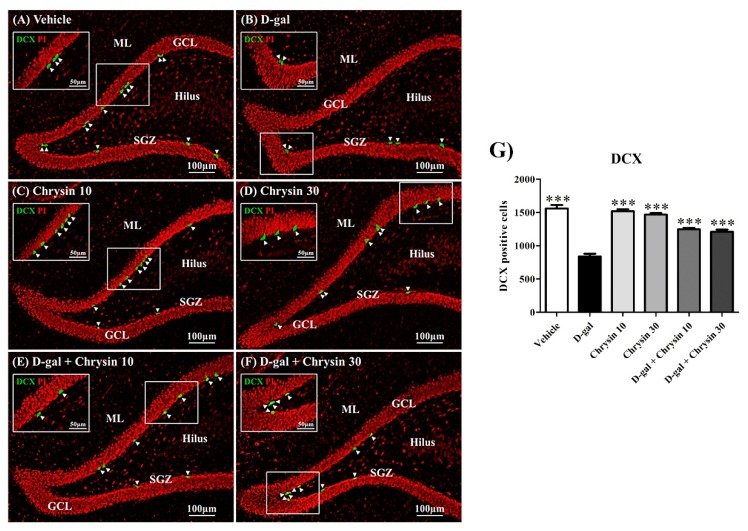
Doublecortin (DCX) immunofluorescence staining of all groups (**A**–**F**). DCX positive cells (green) are indicated by arrowheads, and all nuclei were counterstained with propidium iodide (red). The vehicle, chrysin 10, chrysin 30, and D-gal + chrysin groups had significantly greater numbers of DCX-positive cells than the D-gal group (*** *p* < 0.001, (**G**)). SGZ: subgranular zone; GCL: granule cell layer; ML: molecular layer.

**Table 1 nutrients-12-01100-t001:** Distance moved and velocity (mean ± SEM) in the novel object location (NOL) test after treatment.

Groups	Distance Moved (cm)	Velocity (cm/s)
Vehicle	3018 ± 593.3	1.680 ± 0.3302
D-gal	3001 ± 300.6	1.667 ± 0.1668
Chrysin 10	2404 ± 1337.0	1.337 ± 0.2804
Chrysin 30	2045 ± 505.3	1.138 ± 0.1009
D-gal + chrysin 10	2607 ± 180.8	1.449 ± 0.1807
D-gal + chrysin 30	1688 ± 324.8	0.939 ± 0.1851

**Table 2 nutrients-12-01100-t002:** Distance moved and velocity (mean ± SEM) in the novel object recognition (NOR) test.

Groups	Distance Moved (cm)	Velocity (cm/s)
Vehicle	2273 ± 295.5	1.264 ± 0.1638
D-gal	1800 ± 365.5	1.002 ± 0.2036
Chrysin 10	2495 ± 312.9	1.389 ± 0.1741
Chrysin 30	1590 ± 260.5	0.883 ± 0.1452
D-gal + chrysin 10	2248 ± 469.6	1.250 ± 0.2600
D-gal + chrysin 30	1718 ± 469.2	0.957 ± 0.2609

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
