# Peer review of "Chrysin Protects against Memory and Hippocampal Neurogenesis Depletion in D-Galactose-Induced Aging in Rats"

_nutrients, 2020, doi:10.3390/nu12041100_

Round 1

Reviewer 1 Report

Manuscript ID: nutrients-763555

Manuscript title: Chrysin protects against memory and hippocampal neurogenesis depletion in D-gal-induced aging in rats

Authors investigated the effects of Chrysin on D-galactose-induced aging based on behavioral and morphological studies. Results are interesting, but there are some concerns to consider.

In figure 6, Authors administered BrdU at the first, second, and third day of drug administration and showed the immunohistochemical data for BrdU. However, I wonder BrdU positive cells are only located in the subgranular zone of dentate gyrus even though BrdU integration was done 8 weeks ago. In addition, authors should conduct the double immunohistochemical staining for BrdU and NeuN to confirm the integration into mature neurons.

In figure 7, authors showed the DCX immunoreactive neuroblasts in the dentate gyrus. Where are the dendrites of DCX-immunoreactive neuroblasts? In addition, authors observed 1,500 DCX positive cells in the dentate gyrus, but it is few because DCX is expressed in the immature cells aged 1 to 28 days after birth.

In the introduction and discussion, authors should describe the biological effects of chrysin and compare the results.

Authors should describe how they obtained the Chrysin used in this study.

Author Response

Manuscript ID: nutrients-763555

Manuscript title: Chrysin protects against memory and hippocampal neurogenesis depletion in D-gal-induced aging in rats

Reviewer 1

Authors investigated the effects of Chrysin on D-galactose-induced aging based on behavioral and morphological studies. Results are interesting, but there are some concerns to consider.

In figure 6, Authors administered BrdU at the first, second, and third day of drug administration and showed the immunohistochemical data for BrdU. However, I wonder BrdU positive cells are only located in the subgranular zone of dentate gyrus even though BrdU integration was done 8 weeks ago. In addition, authors should conduct the double immunohistochemical staining for BrdU and NeuN to confirm the integration into mature neurons.

In the present study, we aimed to study the effect of chrysin and brain aging on neurogenesis by determining cell proliferation, cell survival and immature neurons in the subgranular zone of the hippocampal dentate gyrus. After cell proliferation, newly generated cells undergo the differentiation process to be either neurons or glial cells. During this period, we focused on the number of cell survival and immature neurons using BrdU and doublecortin (DCX) staining, respectively. According to our previous data, we mainly found BrdU positive cells in the subgranular zone of the hippocampal dentate gyrus. Only a few of the BrdU positive cells were faintly detected in the granular cell layer. A previous study has also demonstrated BrdU positive cells detected in the subgranular zone of the hippocampal dentate gyrus 1 day, 3 days, 7 days, 4 weeks, 3 months, 6 months and 11 months after BrdU injection to investigate cell survival (Kempermann et al., 2003). These data are in agreement with the present study and our previous studies (Aranarochana et al., 2019; Chaisawang et al., 201, Naewla et al., 2019; Sirichoat et al., 2019). With regard to your suggestion, we have considered to performed the double immunohistochemical staining for BrdU and NeuN in our further study to confirm the integration into mature neurons and investigate mechanisms and actions of chrysin and brain aging.    

Aranarochana, A.; Chaisawang, P.; Sirichoat, A.; Pannangrong, W.; Wigmore, P.; Welbat, J.U. Protective effects of melatonin against valproic acid-induced memory impairments and reductions in adult rat hippocampal neurogenesis. Neuroscience 2019, 406, 580–593.

Chaisawang, P.; Sirichoat, A.; Chaijaroonkhanarak, W.; Pannangrong, W.; Sripanidkulchai, B.; Wigmore, P.; Welbat, J.U. Asiatic acid protects against cognitive deficits and reductions in cell proliferation and survival in the rat hippocampus caused by 5-fluorouracil chemotherapy. PLoS ONE 2017, 12, e0180650.

Kempermann, G.; Gast, D.; Kronenberg, G.; Yamaguchi, M.; Gage, F.H. Early determination and long-term persistence of adult-generated new neurons in the hippocampus of mice. Development 2003, 130, 391.

Naewla, S.; Sirichoat, A.; Pannangrong, W.; Chaisawang, P.; Wigmore, P.; Welbat, J.U. Hesperidin Alleviates Methotrexate-Induced Memory Deficits via Hippocampal Neurogenesis in Adult Rats. Nutrients 2019, 11, 936.

Sirichoat, A.; Krutsri, S.; Suwannakot, K.; Aranarochana, A.; Chaisawang, P.; Pannangrong, W.; Wigmore, P.; Welbat, J.U. Melatonin protects against methotrexate-induced memory deficit and hippocampal neurogenesis impairment in a rat model. Biochem. Pharmacol. 2019, 163, 225233.

In figure 7, authors showed the DCX immunoreactive neuroblasts in the dentate gyrus. Where are the dendrites of DCX-immunoreactive neuroblasts? In addition, authors observed 1,500 DCX positive cells in the dentate gyrus, but it is few because DCX is expressed in the immature cells aged 1 to 28 days after birth.

DCX is a protein marker of immature neurons and DCX expression is highest at the second week of neurogenesis. However, DCX is downregulated when the immature neurons are completely differentiated to mature neurons (Couillard-Despres et al., 2005). In the present study, we aimed to investigate the immature neurons that are generated throughout the lifespan. Counting of immature neurons is estimated by the number of DCX-positive cells multiply 8, which can be a representative of the immature neurons through the whole length of the dentate gyrus (Huang and Herbert, 2006; Umka et al., 2010; Aranarochana et al., 2019; Naewla et al., 2019; Sirichoat et al., 2019). In some studies, the number of DCX positive cells were approximately 1,500 cells, which is similar to the present study (Aranarochana et al., 2019; Naewla et al., 2019; Sirichoat et al., 2019). Another study has revealed that the mean of DCX positive cells was approximately 20 cells using Optimas 6.5 software (CyberMetrics) (Yoo et al., 2011). Taken together, the number of DCX positive cells can be different from our study because of the differences of animal strains, staining and qualification protocols.

As we used a fluorescence microscope (Nikon ECLIPSE 80i, USA) to investigate DCX positive cells, the resolution of the cell is not as good as a confocal microscopy. The dendrites of DCX positive cells were seldom detected as shown at the blue arrowheads in figure 7.

Aranarochana, A.; Chaisawang, P.; Sirichoat, A.; Pannangrong, W.; Wigmore, P.; Welbat, J.U. Protective effects of melatonin against valproic acid-induced memory impairments and reductions in adult rat hippocampal neurogenesis. Neuroscience 2019, 406, 580593.

Couillard-Despres, S.; Winner, B.; Schaubeck, S.; Aigner, R.; Vroemen, M.; Weidner, N.; Bogdahn, U.; Winkler, J.; Kuhn, G.; Aigner, L. Doublecortin expression in adult brain reflect neurogenesis. Eur. J. Neurosci. 2005, 21, 114.

Huang, G.-J.; Herbert, J. Stimulation of Neurogenesis in the Hippocampus of the Adult Rat by Fluoxetine Requires Rhythmic Change in Corticosterone. Biological Psychiatry 2006, 59, 619624.

Naewla, S.; Sirichoat, A.; Pannangrong, W.; Chaisawang, P.; Wigmore, P.; Welbat, J.U. Hesperidin Alleviates Methotrexate-Induced Memory Deficits via Hippocampal Neurogenesis in Adult Rats. Nutrients 2019, 11, 936.

Sirichoat, A.; Krutsri, S.; Suwannakot, K.; Aranarochana, A.; Chaisawang, P.; Pannangrong, W.; Wigmore, P.; Welbat, J.U. Melatonin protects against methotrexate-induced memory deficit and hippocampal neurogenesis impairment in a rat model. Biochem. Pharmacol. 2019, 163, 225233.

Umka, J.; Mustafa, S.; ElBeltagy, M.; Thorpe, A.; Latif, L.; Bennett, G.; Wigmore, P.M. Valproic acid reduces spatial working memory and cell proliferation in the hippocampus. Neuroscience 2010, 166, 1522.

Yoo, D.Y.; Kim, W.; Lee, C.H.; Shin, B.N.; Nam, S.M.; Choi, J.H.; Won, M.-H.; Yoon, Y.S.; Hwang, I.K. Melatonin improves d-galactose-induced aging effects on behavior, neurogenesis, and lipid peroxidation in the mouse dentate gyrus via increasing pCREB expression: Effects of melatonin ond-galactose-induced hippocampal functions. J. Pineal Res. 2012, 52, 21–28.

In the introduction and discussion, authors should describe the biological effects of chrysin and compare the results.

In the introduction we have added about chrysin Chrysin has the ability to diminish apoptosis and memory deficits caused by traumatic brain injury (Rashno et al., 2019).” on page 2, line 57-58, and “In a rat model, it has been reported that chrysin has high potential to protect against 3-Nitropropionic acid-induced mitochondria damage, oxidative stress, neuronal cell death, motor and cognitive impairments (Thangarajan et al., 2016). Chrysin also decreases impairments of the motor behavior, reduction of nigrostriatal dopaminergic neurons in retonone-induced Parkinson in rats (Ahmed et al., 2018).” on page 2, line 62-66.

In the discussion on page 11, line 321-326, we have added about chrysin “Several studies have proved that antioxidant and anti-inflammatory properties of chrysin are able to protect against neuronal cell damage and cognitive impairments. These properties would be a powerful effect of chrysin to ameliorate reduction in neurogenesis in aging rats caused by D-gal. Therefore, the mechanism of action of chrysin in aging rats including antioxidants and neurotrophic factors should be determined to confirm the biological effects of chrysin.”.

Ahmed, M. Neuroprotective role of chrysin in attenuating loss of dopaminergic neurons and improving motor, learning and memory functions in rats; 2018, 12, 35–43.

Rashno, M.; Sarkaki, A.; Farbood, Y.; Rashno, M.; Khorsandi, L.; Naseri, M.K.G.; Dianat, M. Therapeutic effects of chrysin in a rat model of traumatic brain injury: A behavioral, biochemical, and histological study. Life Sciences 2019, 228, 285294.

Thangarajan, S.; Ramachandran, S.; Krishnamurthy, P. Chrysin exerts neuroprotective effects against 3-Nitropropionic acid induced behavioral despairMitochondrial dysfunction and striatal apoptosis via upregulating Bcl-2 gene and downregulating BaxBad genes in male wistar rats. Biomedicine & Pharmacotherapy 2016, 84.

Authors should describe how they obtained the Chrysin used in this study.

We have added chrysin, which is 97% purity determined by HPLC analysis, and purchased from Sigma Aldrich, USA,” on page 2, line 85-86.

Reviewer 2 Report

My questions were answered correctly. Then my opinion is accept the work to publish

Author Response

Reviewer 2

Thank you for your kindness to review and give beneficial suggestions.

Best Regards,

Jariya Umka Welbat

Round 2

Reviewer 1 Report

Manuscript has been improved and I have no further comments. 

This manuscript is a resubmission of an earlier submission. The following is a list of the peer review reports and author responses from that submission.

Round 1

Reviewer 1 Report

Manuscript ID: nutrients-750738

Manuscript title: Chrysin protects against memory and hippocampal neurogenesis depletion in D-gal-induced aging in rats

Authors investigated the effects of Chrysin on D-galactose-induced aging based on behavioral and morphological studies. Results are interesting, but there are some concerns to consider for publication.

In figure 6, Authors administered BrdU at the first, second, and third day of drug administration and showed the immunohistochemical data for BrdU. However, I wonder BrdU positive cells are only located in the subgranular zone of dentate gyrus even though BrdU integration was done 8 weeks ago. In addition, authors should conduct the double immunohistochemical staining for BrdU and NeuN to confirm the integration into mature neurons.

In figure 7, authors showed the DCX immunoreactive neuroblasts in the dentate gyrus. Where are the dendrites of DCX-immunoreactive neuroblasts? In addition, authors observed 1,500 DCX positive cells in the dentate gyrus, but it is few because DCX is expressed in the immature cells aged 1 to 28 days after birth.

In the introduction and discussion, authors should describe the biological effects of chrysin and compare the results.

Authors should describe how they obtained the Chrysin used in this study.

Authors should not use the abbreviation in the title.

Reviewer 2 Report

Authors drawn a very precise experiment to study the beneficial effects of Chrysin on a senescence rat model

The main criticism is that in the present work there are not present an experimental result about the supposed mechanisms of action explaining the protective role of this flavone. Its true that in 2015 Souza et al. demonstrated the well-defined antioxidant properties and BDNF increase induced by Chrysin in aged mice but no references about b-galactosidadse  and Chrysin or other flavone . I suggest to include some simply experiment (Inmunoblot or immunohistochemistry) for some oxidative stress marker or neurotrophic factor increase in this specific model.

 Minor:

Bar charts of IHC figures are not labelled correctly (albeit in the copy I have)